# Self-Acceptance and Interdependence Promote Longevity: Evidence From a 20-year Prospective Cohort Study

**DOI:** 10.3390/ijerph17165980

**Published:** 2020-08-18

**Authors:** Reuben Ng, Heather G. Allore, Becca R. Levy

**Affiliations:** 1Lee Kuan Yew School of Public Policy, National University of Singapore, Singapore 259772, Singapore; 2Lloyds Register Foundation Institute for the Public Understanding of Risk, National University of Singapore, Singapore 117602, Singapore; 3Yale School of Medicine, Yale University, New Haven, CT 06511, USA; heather.allore@yale.edu; 4Yale School of Public Health, Yale University, New Haven, CT 06520, USA; Becca.Levy@yale.edu

**Keywords:** psychological well-being, mortality, psychomics, mediation, preventive health, social gerontology, successful aging

## Abstract

We explored psychosocial pathways to longevity, specifically, the association between psychological well-being and mortality in a 20-year prospective cohort study of 7626 participants. As hypothesized, high self-acceptance and interdependence were associated with decreased mortality risk, controlling for other psychological components (purpose, positive relations, growth, mastery) and potential confounders: personality, depression, self-rated health, smoking status, body mass index (BMI), illness, and demographics. Self-acceptance decreased mortality risk by 19% and added three years of life. Longevity expectation fully mediated the relationship between self-acceptance and mortality. Interdependence decreased mortality risk by 17% and added two years of life. Serenity towards death fully mediated the relationship between interdependence and mortality. This is the first known study to investigate self-acceptance, interdependence, and serenity toward death as promoters of longevity, and distilled the relative contributions of these factors, controlling for covariates—all of which were measured over multiple time points. Theoretically, this study suggests that components of well-being may make meaningful contributions to longevity, and practically recommend that self-acceptance and interdependence could be added to interventions to promote aging health.

## 1. Introduction

The link between psychological well-being and health has received considerable empirical support in recent years. Systematic reviews concluded that positive psychological well-being decreased mortality risk for healthy populations, with the link being stronger in old age [1].

Despite considerable progress in the well-being and health literature, there are several shortcomings: One, over-simplification of psychological well-being by using only a single indicator. Two, psychological (e.g., personality) and physical (e.g., smoking status) covariates are potential confounders that have not been statistically controlled for [2]. Three, well-being factors and covariates have been measured at baseline without subsequent updates. Therefore, important changes that impact longevity have not yet been modeled. 

With the maturation of America’s oldest running cohort study, the Wisconsin Longitudinal Study [3], we aimed to overcome these weaknesses to explore how six psychological well-being dimensions impact longevity after 20 years of follow-up. We also control for psychological (personality factors, depression, self-rated health) and physical (smoking status, body mass index (BMI), illnesses) covariates measured over two time points. This is the first known study to elucidate which well-being factors impact longevity independent of psychological and physical covariates. In addition, we investigated the mediators of the well-being–longevity link. 

Our study is predicated on and seeks to advance Ryff’s psychological well-being theory [4] in relation to longevity. Integrating concepts from Rogers’ fully functioning person to Erikson’s psychosocial development, Ryff proposed six dimensions for positive psychological functioning: self-acceptance (positive attitudes towards self), purpose in life (one’s meaning in life and goals), positive relations (ability to develop trusting relationships), personal growth (willingness to embrace challenging and novel experiences), environmental mastery (feeling of being in charge of one’s life), and autonomy (sense of independence in one’s thoughts and action). The validated scale [5] presents a broad conceptualization of well-being, and researchers have used the positive and negative forms of its components [6]. For example, instead of autonomy, which is associated with increased narrow-mindedness, rigidity in group settings, and poorer functioning, researchers use the opposite construct of interdependence that may present benefits in old age [7,8]. In any case, no known studies have looked at all six dimensions in relation to longevity, and our study aimed to close this gap. 

Of broader significance, we extended well-being theory by asking which aspects promote longevity. Although all aspects are crucial to positive functioning, differences exist across age groups and outcomes. Previous studies suggest that self-acceptance and interdependence are most relevant in old age [9], although none have explored their benefit in relation to longevity. Against this backdrop, we constructed four hypotheses and provided support below: One, self-acceptance is associated with decreased mortality risk, after adjusting for demographics, psychological covariates, and physical covariates. Two, longevity expectation mediates the acceptance–morality link. Three, interdependence is associated with decreased mortality risk, after adjusting for demographics, psychological covariates, and physical covariates. Four, serenity towards death mediates the interdependence–mortality relation. In the following sections, we provide support for each hypothesized model.

### 1.1. Self-Acceptance, Longevity Expectation, Longevity

In hypothesizing that the self-acceptance–longevity relationship is mediated by longevity expectation, we tested, for the first time, all three variables in a single model. Previous cross-sectional studies show that higher self-acceptance is associated with decreased frailty [10], and better mental health [11] but none investigated the link between self-acceptance and mortality. Given the physical and psychological benefits of self-acceptance, we conjected that higher self-acceptance will be associated with decreased mortality risk.

We hypothesized that this pathway is mediated by longevity expectation. Older adults who are more accepting of changes, brought about by age, are expected to live longer [12]. Perhaps, elders who are more self-accepting develop greater resilience against life’s vicissitudes and deepen one’s will to live [2]. A 23-year longitudinal study of participants in Ohio found that one’s will to live was an important pathway via which positive age stereotypes decreased mortality risk [13]. Taken together, we proposed that participants with higher self-acceptance would report higher longevity expectations.

It has also been shown, intriguingly, that one’s longevity expectation predicts one’s mortality. At the ecological level, mortality among Chinese dipped before important festivals (e.g., Harvest Moon) and peak thereafter [14]. For near-death Chinese elders, these festivals are especially treasured as they bring the family together, and it is fair to assume that they expect to live through it. The impact of longevity expectation on mortality is also evident at the individual level. Findings from national cohort studies such as the Health and Retirement Survey (HRS) show that individuals who expect to live longer actually do [15]; they tend to exercise more and engage in healthy behaviors [16,17]. Given that bivariate associations were found between acceptance, longevity expectation, and mortality, we went one step further to propose a pathway model where longevity expectation mediates the acceptance–mortality link. 

### 1.2. Interdependence, Serenity towards Death, Longevity

We hypothesize that the link between interdependence and longevity is mediated by serenity towards death. While autonomy is found to be beneficial for goal pursuit in younger adults, increased autonomy results in narrow-mindedness and rigidity during old age [18]. High self-reported autonomy in old age is characterized by stubbornness and associated with biases, appraisal shortfalls, and even poorer functioning [19,20]. A related study [21] investigated the effect of autonomy and interdependence on physical functioning in the USA (high autonomy) and Japan (high interdependence), and found that interdependence attenuated the negative effects of age stereotypes on physical functioning. Given its benefits, we predicted that interdependence would be protective against mortality risk.

We further hypothesized that the interdependence–longevity link would be mediated by serenity towards death. Most studies on serenity towards death are qualitative. Individuals higher in interdependence are more likely to adopt a “letting things go” attitude and report higher serenity towards death [22]. In a spirituality intervention, patients in palliative care were taught to be less self-focused about what death meant. They reported increased serenity and acceptance of death [23]. Interviews with survivors of cancer found that being other-focused or interdependent is a key antecedent of serenity towards death [24]. These studies show, albeit qualitatively, that interdependence is linked to serenity towards death, and our study is the first to test this relationship quantitatively. 

Although the link between serenity towards death and mortality has not been studied, several studies found that serenity is associated with positive factors that may lower mortality risk. In near-death situations, patients and their families who were counseled to adopt a Buddhist approach of serenity made decisions that improved the patient’s quality of life [25]. Kocher (1994) found that participants primed with a peaceful death were more likely to report a greater commitment to exercising and other self-preserving behaviors [26]. These studies set up a mediational model of interdependence–serenity–mortality that has not been tested before. Further, our study is unprecedented in two ways: a quantitative study of serenity toward death in a realm that has been mostly qualitative, and introducing interdependence and serenity toward death as factors associated with longevity.

### 1.3. Potential Confounders

Apart from known confounders—demographics, depression, self-rated health, smoking, and BMI—we controlled for potential factors that were previously not considered in psychosocial models of longevity—personality and other domains of psychological well-being. Personality factors are known to impact mortality. A follow-up with Terman’s cohort of gifted children after 67 years found that neuroticism is associated with higher mortality risk for females but lower risk for males [27]. A study of the oldest old found that neuroticism and agreeableness are protective against mortality [28]. Another sample of all-male clergy found opposite results—mortality risk was nearly two times higher for those with high neuroticism [29]. These differing findings hint that other psychological factors may be involved but this has not been studied in terms of personality. 

Domains of Ryff’s psychological well-being are mostly studied independently without controlling for the other domains. Using a different definition and measurement of mastery, Surtees and colleagues found an association with decreased depression and mortality [30]. Mastery also mediated the social support–physical health relationship [31]. Sense of purpose was associated with decreased mortality for heart transplant patients [32,33]. Positive relations have not been researched with mortality although related constructs such as increased social support are linked to better physical health [34]. Personal growth is the most under-researched among well-being components, with several studies on pediatric nurses [35] but none on older adults. Given that other domains of psychological well-being are associated with health outcomes, and could potentially impact mortality, we controlled for them in our study.

### 1.4. Present Study

We investigated the impact of self-acceptance and interdependence on mortality risk in a 20-year prospective cohort study, controlling for potential confounders including the five factors of personality, four dimensions of well-being, depression, self-rated health, physical factors (smoking, BMI, illness), and demographics. There are four hypotheses.

**Hypothesis** **1:***Self-acceptance is associated with decreased mortality risk, after adjusting for covariates*.

**Hypothesis** **2:***Interdependence is associated with decreased mortality risk, after adjusting for similar covariates*.

**Hypothesis** **3:***Longevity expectation mediates the acceptance–morality link*.

**Hypothesis** **4:***Serenity towards death mediates the interdependence–mortality relationship*.

## 2. Materials and Methods

### 2.1. Data and Participants

The Wisconsin Longitudinal Study (WLS) is the longest running cohort study in the United States, consisting of 10,317 participants who graduated from Wisconsin high schools in 1957 [36]. One in three graduates were randomly selected to take part in the WLS and were followed up for 50 years, with over 90% retention rates among those alive during each wave (1975, 1992, 2003). 

Psychological well-being measures were introduced in 1992. Thus, we included WLS participants who were alive in 1992 and completed the psychological well-being modules (*N* = 7626). Included and excluded participants did not differ significantly across gender (*p* = 0.08), race (*p* = 0.40), marital status (*p* = 0.09), educational status (*p* = 0.12), and age (*p* = 0.31). Among included participants, 53.1% were female, 99.76% were white, 82.75% were married, and 47.85% went to college, and the participants had a mean age of 53.14 (SD = 0.5) in 1992. These demographics are reflective of the population in Wisconsin. 

### 2.2. Measures 

Independent variables: self-acceptance and interdependence measured in 1992 and 2003. Both variables were measured using Ryff’s Psychological Well-being Scale that has been validated in numerous studies worldwide [37,38]. Responses are measured from 1 (agree strongly) to 6 (disagree strongly), such that lower scores denote better well-being. We reversed all scores such that higher scores reflected better well-being, specifically, higher scores represented higher self-acceptance and interdependence. We used the mean scores for each component in our analyses. To enable Kaplan–Meier analyses, we created dichotomous versions through a median split [1].

Three self-acceptance questions are “I like most parts of my personality”; “when I look at the story of my life, I am pleased how things have turned out”; and “in many ways, I feel disappointed about my achievements in life.” Interdependence items are “I judge myself by what I think is important, not by the values of what others think is important”; “I tend to be influenced by people with strong opinions”; “I stick to my opinions even if they are contrary to the general consensus”. These questions, which measured autonomy, were reversed scored to represent interdependence. The subscales were reliable, with alpha coefficients between 0.75 and 0.89 in our sample. 

Outcome. Date of death was obtained from the U.S. National Death Index, a compilation by the National Center for Health Statistics of all death-record information by state-level vital statistics offices. We used the number of days participants survived from the baseline interview in 1992 to December 31 2012. 

Demographic covariates. Baseline characteristics were age, gender, education, race, and marital status. Education was measured by whether the participant attended college and marital status was classified into married, single, or divorced/separated/widowed.

Psychological covariates measured in 1992 and 2003. Personality was measured by the Big Five Inventory with 44 items [39]. Participants rated themselves with respect to each statement from 1 (strongly disagree) to 5 (strongly agree). Scores were summed with higher scores reflecting higher levels of each personality factor. Questions include “Is full of energy” (extraversion), “Is generally trusting” (agreeableness), “does a thorough job” (conscientiousness), “can be tense” (neuroticism), and “has an active imagination” (openness). All sub-scales and respective Cronbach’s alphas in the current sample: extraversion (α = 0.73), agreeableness (α = 0.73), conscientiousness (α = 0.73), neuroticism (α = 0.74), and openness (α = 0.67).

Sample questions for four other well-being dimensions: purpose in life, “Some people wander aimlessly through life, but I am not one of them”; positive relations, “People would describe me as a giving person, willing to share my time with others”; personal growth, “For me, life has been a continuous process of learning, changing”; environment mastery, “In general, I feel I am in charge of the situation in which I live”.

Self-rated health was measured by “How would you rate your health at the present time”, rated from 1 (very poor) to 5 (excellent). Depression was measured by the number of depressive symptoms that lasted for 2 weeks or more through six yes/no statements. A sample question: “Did you have two weeks or more when you lacked energy or felt tired all the time, even when you had not been working very hard?”

Physical covariates: smoking, BMI, illness count measured in 1992 and 2003. Smoking was measured by “do you smoke now”, a yes/no question, and body mass index (BMI) was calculated using the American formula: (weight in pounds x 703)/(height in inches x height in inches). Participants were also asked if they were diagnosed (yes/no) with any of the following 16 medical conditions: anemia, asthma, arthritis or rheumatism, bronchitis or emphysema, cancer, chronic liver trouble, diabetes, serious back trouble, heart trouble, high blood pressure, circulation problems, kidney or bladder problems, ulcer, allergies, multiple sclerosis, colitis. All positive answers were summed to form the illness count covariate. 

Mediating variables (MV) measured in 2003. Longevity expectation was measured by “what are the chances that you will live for another 20 years” on a scale of 0 *(no chance)* to 10 *(absolutely certain)*. Higher scores indicate higher longevity expectations. Serenity toward death was measured by “I would neither fear death nor welcome it” on a scale of 1 *(disagree strongly)* to 6 *(agree strongly)*. Higher scores correspond with higher serenity.

### 2.3. Analytic Strategy

We conducted our analysis in three stages. First, we used AMOS 18 (AMOS, Chicago, USA) [40] to run a confirmatory factor analysis (CFA) to establish the factor structure of well-being in both 1992 (wave 1) and 2003 (wave 2). This step was paramount, given differing findings in other studies [38,41]. Multiple indices were used to evaluate model fit [42] in our CFA: root mean square error approximation (RMSEA) [43] that measures discrepancy per degree of freedom and imposes a penalty for adding complexity to a model without substantially improving model fit. Smaller RMSEA values indicate better model fit, with values less than 0.05 indicating a “close fit”, between 0.05 and 0.08 corresponding to an “acceptable” fit, and RMSEA values larger than 0.10 suggesting a “poor fit” [44]. The comparative fit index (CFI) and the Tucker–Lewis index (TLI) measure the relative reduction in model misfit when comparing the target model relative to a baseline (independence) model. The CFI and TLI values greater than 0.90 have been considered an indication of an acceptable fit of the model to the observed data. 

Second, we used the dimensions of psychological well-being (acceptance and interdependence), distilled from the CFA, as predictors in the Cox regression models. Of note, this is the first known study to use time-dependent predictors and covariates in studies linking psychological well-being and mortality. Such models are advantageous because they take into account changes in psychological well-being, physical covariates, and psychological covariates between 1992 and 2003 [45]. We ran four models: Model 1 investigates the associations between well-being and all-cause mortality over 20 years; Model 2 adds demographic covariates measured at baseline in Model 2; Model 3 adds time-dependent psychological covariates (personality, self-rated health, depression); Model 4 is the full model that includes time-dependent physical covariates (smoking, BMI, illness count). We used−2 log-likelihood statistics (−2LL) to determine how our Cox regression models fit our data. Lastly, Kaplan–Meier survival curves were calculated to assess differences in survival between the components of psychological well-being that reached significance in Model 4. A significance level of 0.05 was chosen a priori. 

Mediation occurs when (1) the Independent Variable (IV) significantly predicts the outcome, (2) the IV significantly predicts the Mediating Variable (MV), and (3) the IV–outcome link is significantly reduced (also known as statistical suppression effect) when the MV is entered after the IV in a hierarchical regression [46,47]. The indirect effect of the IV through the MV is the product of the coefficients of the slope of the MV in the Step (2) and (3) regressions. To be acceptable as a mediator, the indirect effect has to be significantly greater than zero. In particular, the 95% CIs around the indirect effect from numerous bootstrap re-samples should exclude zero [48]. The PRODCLIN program (IBM, Armonk, NY, USA) [46] was used to calculate the 95% CI. Full mediation occurs when, in the hierarchical regression (3), the MV is significant while the IV is not. This means that the effect of the IV on the outcome takes place through the MV [49,50]. On the other hand, partial mediation happens when the MV reaches significance along with the significance of the IV, albeit diminished, in the hierarchical regression (3). This would mean that the effect of the IV on the outcome goes partially through the MV (for a review, see MacKinnon, Fairchild, and Fritz, 2007). A significance level of 0.05 was chosen a priori. 

## 3. Results 

### 3.1. Confirmatory Factor Analysis

Consistent with numerous studies [37], our data achieved good fit with the six-factor structure of psychological well-being in 1992 (wave 1), χ^2^ = 77.36, CFI = 0.97, TLI = 0.92, RMSEA = 0.09, and 2003 (wave 2), χ^2^ = 47.41, CFI = 0.98, TLI = 0.95, RMSEA = 0.07. Of the six factors, two are predictors (acceptance and interdependence) and four are covariates (purpose, relations, growth, mastery).

### 3.2. Kaplan–Meier Analysis

We performed a Kaplan–Meier analysis to examine the difference in survival between groups that were high and low in self-acceptance and interdependence. Consistent with previous studies and recommendations [1], individuals above median self-acceptance scores were categorized as high, while the rest were grouped as low. Participants high in self-acceptance had a survival benefit of 3 years relative to those with low self-acceptance (log-rank test: χ^2^ = 8.58, *p* = 0.003) at the 90th percentile (see Figure 1). The same pattern emerged for interdependence: those high in interdependence evidenced a survival benefit of 2 years compared to those low in interdependence (log-rank test: χ^2^ = 4.59, *p* = 0.032) at the 90th percentile (see Figure 2).

### 3.3. Cox Regression with Time-Varying Covariates

As predicted, in Model 4, high self-acceptance was found to be protective against mortality risk (hazard ratio = 0.81, *p* = 0.01). Every one-unit increase in self-acceptance decreased mortality risk by 19% after controlling for demographics, psychological covariates, and physical covariates. Likewise, high self-reported interdependence was associated with lower mortality risk (hazard ratio = 0.83, *p* = 0.03). Specifically, a one-unit increase in interdependence lowered mortality risk by 17% after controlling for all covariates. Other components of psychological well-being did not reach significance (Table 1). To address potential concerns for collinearity among well-being components, we ran six separate models, controlling for similar covariates, and did not find evidence for collinearity (e.g., variance inflation). These model outputs are available on request.

We found that self-acceptance and interdependence decreased mortality risk, supporting hypotheses 1 and 2. The protective effect of these factors persisted even after controlling for other well-being dimensions, personality, depression, and other physical covariates (smoking, BMI, illness), suggesting that psychological pathways may have been involved. 

### 3.4. Mediational Analysis: Acceptance–Longevity Expectation–Mortality

Figure 3 presents the standardized coefficients (β) that supported full mediation. (1) Self-acceptance significantly predicted mortality, β = −0.16 (95% CI: −0.23, −0.090), *p* < 0.0001. (2) Self-acceptance significantly predicted longevity expectation, β = 0.59 (95% CI: 0.50, 0.68), *p* < 0.0001. (3) When longevity expectation was entered after self-acceptance in the hierarchical regression, longevity expectation (MV) was significant, β = −0.13 (95% CI: −0.016, −0.10), *p* < 0.0001, while self-acceptance (IV) became non-significant, β = −0.064 (95% CI: −0.17, 0.043), *p* = 0.24. The 95% CI from 5000 bootstraps excluded zero (−0.089 to −0.054). This means that self-acceptance decreased mortality risk via increased longevity expectation, supporting Hypothesis 3.

### 3.5. Mediational Analysis: Interdependence–Serenity–Mortality

Likewise, full mediation was established (Figure 4). (1) Interdependence significantly predicted mortality, β = −0.091 (95% CI: −0.16, −0.012), *p* = 0.023. (2) Interdependence significantly predicted serenity toward death, β = 0.20 (95% CI: 0.15, 0.24), *p* = 0.024. (3) When serenity was entered after interdependence in the hierarchical regression, serenity (MV) was significant, β = 0.07 (95% CI: 0.0060, 0.13), *p* = 0.033, while interdependence (IV) became non-significant, β = 0.015 (95% CI: −0.10, 0.13), *p* = 0.90. The 95% CI from 5000 bootstraps excluded zero (0.003 to 0.026). This meant that interdependence decreased mortality risk via increased serenity toward death, supporting Hypothesis 4.

## 4. Discussion

The purpose of our study was to investigate the impact of self-acceptance and interdependence on longevity over 20 years, controlling for other well-being dimensions (purpose, relations, growth, mastery) and potential confounders such as personality, depression, self-rated health, smoking status, BMI, illness, and demographics. After adjusting for these covariates, self-acceptance and interdependence remained protective against mortality risk. Specifically, self-acceptance decreased mortality risk by 19% and interdependence by 17%. Participants high in self-acceptance and interdependence evidenced a 3-year and 2-year survival advantage, respectively. We found that longevity expectation fully mediated the acceptance–mortality link, and serenity towards death fully mediated the interdependence–mortality relation.

### 4.1. Theoretical Implications 

Our project makes three theoretical contributions. The first is elucidating the psychological pathways to longevity. Previous studies concentrated on mediators of the positive age stereotypes and longevity: will to live [13], expectations of better health [51], and self-efficacy [52]. We uncovered two new pathways: self-acceptance–longevity expectation–longevity, and interdependence–serenity toward death–longevity. Our findings are predicated on prior studies that found respective bivariate IV–MV relationships but lacked the data to test mediational models with mortality as outcome. These psychological pathways are plausible because they promote self-efficacy and health behaviors that are protective. For example, elders with greater longevity expectations tend to adopt healthier lifestyles and exercise more [17]; higher serenity towards death is associated with higher quality of life [25]. Future studies should extend the pathways to test such three-step mediation hypotheses.

Second, we controlled for other domains of psychological well-being and still found that self-acceptance and interdependence predicted longevity, supporting previous suggestions that these two factors are most pertinent in old age [9]. Essentially, we extended well-being theory by showing that acceptance and interdependence are most relevant in promoting longevity. However, this does not imply that other domains are unimportant; with respect to longevity only, self-acceptance and interdependence stood out as key predictors. These results cannot be explained by collinearity. Intuitively, we speculate that *purpose* and *growth*, while essential for goal pursuit at youth, are not as important during one’s twilight years. This conjecture is supported by findings that growth and purpose decreased with age [53]. Similarly, older adults have reached a point of stability in positive relations and environmental mastery that they may no longer be relevant to longevity. The norms corroborate with our speculations—when compared to young people [54], older adults in our sample score higher in positive relations and mastery. The relevance of well-being domains across the lifespan would be an intriguing topic for future research. 

Third, we distilled relative contributions of psychological well-being and personality in predicting longevity. Separate studies show that well-being promotes longevity [1] and certain aspects of personality predict mortality [27] but none have studied them together. In our study, well-being components (acceptance and interdependence), but not personality factors, predicted longevity. This suggests that well-being factors have greater influence in promoting longevity than personality, which is good news since well-being is arguably more modifiable than personality [6,55].

### 4.2. Practical Implications

Besides theoretical significance, our study contributes to practical application. Programs that promote well-being, especially self-acceptance and interdependence, can form aspects of non-invasive interventions to meaningfully prolong the lives of healthy seniors. Geriatric counseling can focus on working through one’s sense of self-acceptance and interdependence to increase self-efficacy and coax healthy behaviors. If anything, increasing the elderly’s awareness of such findings through the media will bring across the message that positive mental well-being is as paramount to successful aging as physical health. This study contributes to the evidence base that could be used to convince policymakers that promoting positive psychological well-being is a useful public health strategy for aging societies. 

### 4.3. Limitations and Future Studies

Several shortcomings in this study provide opportunities for future research. Our mediators were measured with single questions and may not have captured the constructs fully. While this is a limitation, it represents one of the first attempts to measure and quantitatively study longevity expectations and serenity toward death as mediators. Moreover, the items were consistent with prior definitions of the respective constructs that were used in prior research [21]. Nevertheless, future studies should take up these measurement issues. Another concern is multi-collinearity given the correlation among components of well-being. We ran six independent models with respective well-being components to check for symptoms of collinearity (e.g., variance inflation) and found no cause for concern. 

Future studies could examine how culture moderates the relationship between self-acceptance/interdependence and longevity. Given that cultural differences influence many curious facets of perceptions [56,57,58,59], it will be interesting to understand how these differences affect psychosocial influences and attitudes [60] on longevity. The WLS would also benefit from another wave of data collection to update the measurements of well-being and delineate its link to mortality risk. Finally, the next step is to apply these findings to design interventions for successful aging in light of historical findings and disability projections [61,62]. 

## 5. Conclusions

In conclusion, staying smoke-free and eating right are not the only strategies to live long and prosper. Successful aging is achieved by both the body and mind. Acceptance of oneself and helping one another are stepping stones in a complicated pathway to longevity. 

## Figures and Tables

**Figure 1 ijerph-17-05980-f001:**
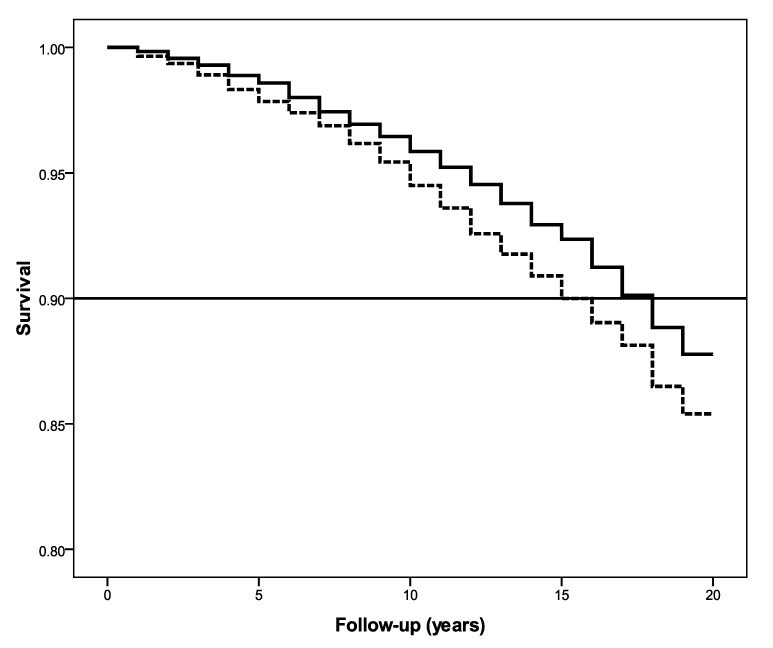
Higher self-acceptance associated with increased survival.

**Figure 2 ijerph-17-05980-f002:**
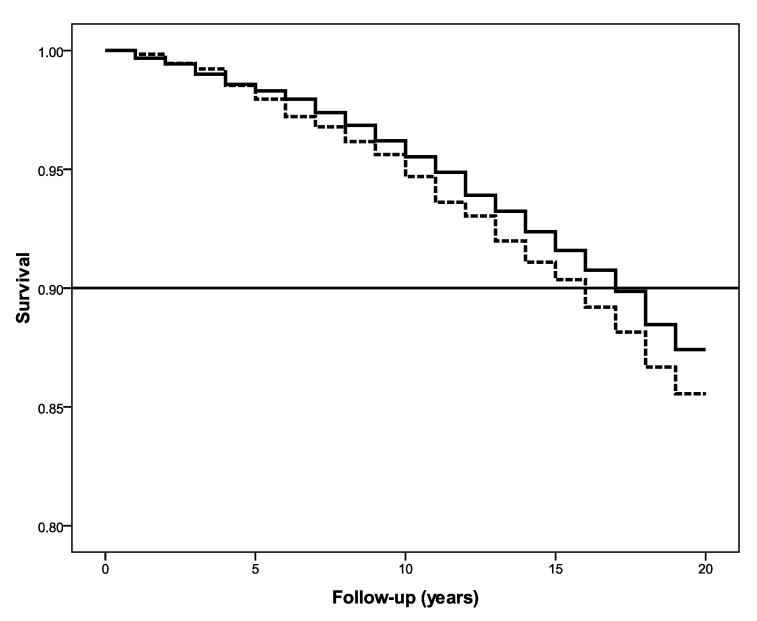
Higher self-reported interdependence associated with increased survival.

**Figure 3 ijerph-17-05980-f003:**
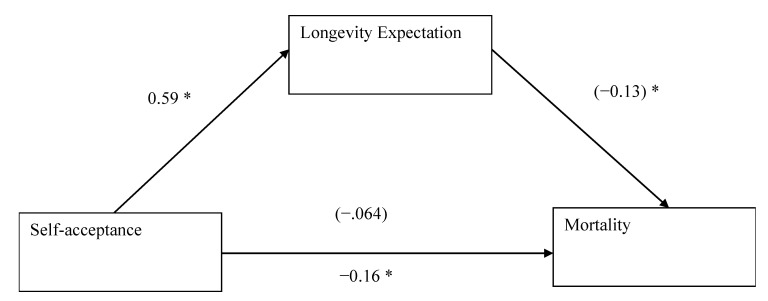
Longevity expectation fully mediated the association between self-acceptance and all-cause mortality (time-to-event). Parentheses denote results of hierarchical regression where longevity expectation was significant while self-acceptance was not. * *p* < 0.05.

**Figure 4 ijerph-17-05980-f004:**
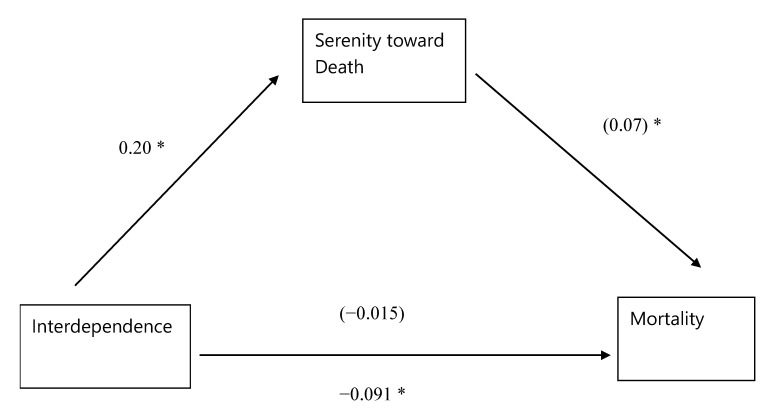
Serenity toward death fully mediated the association between interdependence and all-cause mortality (time-to-event). Parentheses denote results of hierarchical regression where serenity toward death emerged significant while interdependence did not. * *p* < 0.05.

**Table 1 ijerph-17-05980-t001:** Results of multivariable Cox regression model (with time-dependent covariates) for the effect of psychological well-being on all-cause mortality over 20 years.

Psychological Well-Being	Model 4		Model 3		Model 2		Model 1	
	HR ^1^ (95% CI ^2^)	*p*	HR (95% CI)	*p*	HR (95% CI)	*p*	HR (95% CI)	*p*
Self-acceptance	0.81 (0.69, 0.96)	0.015	0.83 (0.71, 0.98)	0.024	0.87 (0.77, 0.99)	0.036	0.88 (0.78, 0.99)	0.039
Sense of purpose	0.89 (0.75, 1.06)	0.19	0.90 (0.78, 1.06)	0.22	0.93 (0.82, 1.06)	0.27	0.90 (0.80, 1.01)	0.081
Positive relations	1.05 (0.91, 1.20)	0.51	1.05 (0.92, 1.19)	0.49	0.98 (0.89, 1.08)	0.70	0.99 (0.91, 1.08)	0.84
Personal growth	1.10 (0.93, 1.30)	0.27	1.09 (0.93, 1.29)	0.28	1.09 (0.96, 1.23)	0.20	1.04 (0.92, 1.17)	0.51
Environmental mastery	1.06 (0.88, 1.28)	0.51	1.07 (0.89, 1.28)	0.47	1.04 (0.90, 1.19)	0.63	1.04 (0.91, 1.18)	0.60
Interdependence	0.83 (0.66, 0.99)	0.029	0.83 (0.66, 0.98)	0.019	0.87 (0.75, 0.98)	0.022	0.86 (0.74, 0.96)	0.008
**Demographics**								
Age	1.20 (1.02, 1.42)	0.028	1.19 (1.01, 1.39)	0.034	1.17 (1.02, 1.33)	0.020		
Gender								
Male	Reference							
Female	0.80 (0.67, 0.96)	0.018	0.76 (0.64, 0.91)	0.0023	0.73 (0.63, 0.83)	<0.001		
Education								
No college	Reference							
Attended college	0.73 (0.60, 0.87)	<0.001	0.74 (0.62, 0.88)	<0.001	0.74 (0.64, 0.85)	<0.001		
Race								
Whites	Reference							
Blacks	3.72 (1.38, 10.01)	0.0093	3.53 (1.31, 9.47)	0.013	2.62 (1.08, 6.31)	0.032		
Marriage								
Married	Reference							
Single	1.81 (1.27, 2.60)	0.0011	1.69 (1.19, 2.40)	0.0036	1.66 (1.24, 2.22)	<0.001		
Divorced/separated	1.85 (1.48, 2.31)	<0.001	1.74 (1.41, 2.16)	<0.001	1.68 (1.41, 2.01)	<0.001		
**Psychological Covariates**								
Openness	0.91 (0.80, 1.04)	0.16	0.94 (0.83, 1.06)	0.30				
Neuroticism	0.97 (0.87, 1.08)	0.54	0.95 (0.86, 1.06)	0.36				
Conscientiousness	0.90 (0.78, 1.04)	0.17	0.88 (0.77, 1.01)	0.07				
Agreeableness	1.02 (0.89, 1.16)	0.80	1.01 (0.89, 1.15)	0.83				
Extroversion	0.98 (0.87, 1.09)	0.68	0.96 (0.86, 1.06)	0.39				
Depression	0.99 (0.94, 1.04)	0.64	0.99 (0.94, 1.04)	0.63				
Self-rated health	0.99 (0.86, 1.14)	0.89	1.01 (0.89, 1.15)	0.89				
**Physical Covariates**								
Smoking								
No	Reference							
Yes	1.00 (0.77, 1.28)	0.99						
Illness count ^3^	1.01 (0.96, 1.08)	0.62						
Body mass index ^4^	0.99 (0.97, 1.01)	0.27						

Notes. ^1^. Hazard ratio. ^2^. Confidence intervals. ^3^. Participants reported if they were diagnosed (yes/no) with any of 16 medical conditions (anemia, asthma, arthritis or rheumatism, bronchitis or emphysema, cancer, chronic liver trouble, diabetes, serious back trouble, heart trouble, high blood pressure, circulation problems, kidney or bladder problems, ulcer, allergies, multiple sclerosis, colitis). All positive answers were summed to form the illness count covariate. ^4^. Body mass index was calculated using the American formula: (weight in pounds x 703)/(height in inches x height in inches).

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
