# Peer review of "Self-Acceptance and Interdependence Promote Longevity: Evidence From a 20-year Prospective Cohort Study"

_ijerph, 2020, doi:10.3390/ijerph17165980_

Round 1

Reviewer 1 Report

Please, see the limitations and strengths of the study bellow:

Strengths:

- The Research Questions are clearly described

- It is clear from Abstract in how far the research hypotheses are supported.

- Good Method section.

- The description of how the data were analyzed is clarity.  

- Nice description why this is a relevant topic and what the theory behind the research questions is.

Limitations:

- References needs to be updated

- The framework and discussion chapter still needs work.

Please, see the suggestion bellow:

INTRODUCTION

- The Research Questions are clearly described

- It is clear from Abstract in how far the research hypotheses are supported.

METHODS

- Nice longitudinal study

- Good Method section.

RESULTS AND DATA INTERPRETATION

- Well-structured

- Results supported by data

DISCUSSION

- References needs to be updated

- Discussion chapter should be re-written.

So, in my opinion, the study is very interesting and worth being published. In fact, the article is well-structured and the results are presented clearly. Nevertheless, I feel I can give the followings suggestions:

  1. The framework should have a more recently studies. Please, include more citations about 2018, 2019, 2020.
  2. I am afraid that the discussion chapter still needs work. I here offer some insight into how the discussion part should be re-written:

- First of all, I suggest to include one short paragraph summarizing the purpose of the study and the main findings obtained.

- Second, I think that the whole discussion needs to be re-structured to include: theoretical implications of the study, practical implications, limitations and future studies and a brief conclusion.

I think the authors can easily follow the suggestions I have given in this review and make a new version of their interesting paper.

All best wishes.

Author Response

Thank you for your important feedback.  Please see the attachment. Thank you.

Reviewer 2 Report

In the manuscript “Self-Acceptance and interdependence promote longevity: evidence from a 20-year prospective cohort study” the authors investigate the impact of self-acceptance and interdependence on mortality risk in a 20- year prospective cohort study, controlling for potential confounders including personality factors, other dimensions of well-being, depression, self-rated health, physical factors and demographics. The manuscript is well written and covers important issues. However, some minor changes should be done before publication:

1. The abstract should be reviewed as there seems to be a small error in the percentage decrease in mortality from self-acceptance. So on page 1, line 18 says: “Self-acceptance decreased mortality risk by 17%”

But on page 8 line 276 say: “Every one-unit increase in self-acceptance decreases mortality risk by 19%”

And on page 12, lines 322,323 say “Specifically, self-acceptance decreased mortality risk by 19%”

2. On page 15, lines 475-476, reference number 36, authors Sewell, W. H.; Hauser, R. M.; Springer, K. W.; Hauser, T. S. the year of publication is missing

3. The limitations should include that data collection has been done for several years. It should not be forgotten that today's society is characterized by a very rapid change both at the social level and in medical and surgical technology and in lifestyles that can influence mortality.

4. References should be updated as there are very few from the last 10 years and none from the last 5 years.

Author Response

(The authors gave the same response as above.)
